# Exploring shared memory architectures for end-to-end gigapixel deep learning

**Lucas W. Remedios**[*1]                          LUCAS.W.REMEDIOS@VANDERBILT.EDU
[1] *Vanderbilt University, Nashville, TN, USA*
**Leon Y. Cai**[*1]                                   LEON.Y.CAI@VANDERBILT.EDU
**Samuel W. Remedios**[2,3]                              SAMUEL.REMEDIOS@JHU.EDU
[2] *Johns Hopkins University, Baltimore, MD, USA*
[3] *National Institutes of Health, Bethesda, MD, USA*
**Karthik Ramadass**[1]                        KARTHIK.RAMADASS@VANDERBILT.EDU
**Aravind Krishnan**[1]                        ARAVIND.R.KRISHNAN@VANDERBILT.EDU
**Ruining Deng**[1]                                   R.DENG@VANDERBILT.EDU
**Can Cui**[1]                                      CAN.CUI.1@VANDERBILT.EDU
**Shunxing Bao**[1]                            SHUNXING.BAO@VANDERBILT.EDU
**Lori A. Coburn**[4,5]                              LORI.COBURN@VUMC.ORG
[4] *Vanderbilt University Medical Center, Nashville, TN, USA*
[5] *Veterans Affairs Tennessee Valley Healthcare System, Nashville, TN, USA*
**Yuankai Huo**[1]                                 YUANKAI.HUO@VANDERBILT.EDU
**Bennett A. Landman**[1]                        BENNETT.LANDMAN@VANDERBILT.EDU

**Editors:** Accepted for publication at MIDL 2023

## Abstract

Deep learning has made great strides in medical imaging, enabled by hardware advances in GPUs. One major constraint for the development of new models has been the saturation of GPU memory resources during training. This is especially true in computational pathology, where images regularly contain more than 1 billion pixels. These pathological images are traditionally divided into small patches to enable deep learning due to hardware limitations. In this work, we explore whether the shared GPU/CPU memory architecture on the M1 Ultra systems-on-a-chip (SoCs) recently released by Apple, Inc. may provide a solution. These affordable systems (less than $5000) provide access to 128 GB of unified memory (Mac Studio with M1 Ultra SoC). As a proof of concept for gigapixel deep learning, we identified tissue from background on gigapixel areas from whole slide images (WSIs). The model was a modified U-Net (4492 parameters) leveraging large kernels and high stride. The M1 Ultra SoC was able to train the model directly on gigapixel images ($16000 \times 64000$ pixels, 1.024 billion pixels) with a batch size of 1 using over 100 GB of unified memory for the process at an average speed of 1 minute and 21 seconds per batch with Tensorflow 2/Keras. As expected, the model converged with a high Dice score of $0.989 \pm 0.005$. Training up until this point took 111 hours and 24 minutes over 4940 steps. Other high RAM GPUs like the NVIDIA A100 (largest commercially accessible at 80 GB, ∼$15000) are not yet widely available (in preview for select regions on Amazon Web Services at $40.96/hour as a group of 8). This study is a promising step towards WSI-wise end-to-end deep learning with prevalent network architectures.

**Keywords:** Gigapixel, GPU, Large Patch, Computational Pathology, Segmentation

---

[*] Contributed equally

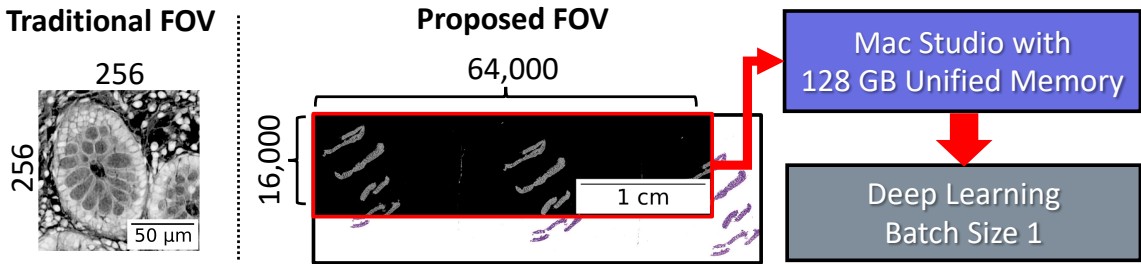

Figure 1: Usually, small patches (eg. $256 \times 256$ pixels) are used in computational pathology. Enabled by the unified memory architecture, we instead use a gigapixel area ($16000 \times 64000$ pixels) from preprocessed images, which is larger than a traditional single tissue field of view.

## 1. Introduction

The deep learning revolution has largely been possible due to GPU acceleration (Dean, 2020). When training on very large images, GPU RAM may not be sufficient for a batch size of 1 (Jain et al., 2020). In this case, model parallelism can be implemented. However, model parallelism requires multiple available GPUs (Yadan et al., 2013) and introduces communication overhead (Keuper and Preundt, 2016).

In computational pathology, gigapixel (1 billion pixels) images are standard (Dimitriou et al., 2019). Instead of training on gigapixel images directly, popular approaches use small patches, for example $256 \times 256$ pixels (Chen et al., 2022). Unfortunately, small patches do not provide global contextual information, thus larger patches are still desired (Chen et al., 2022). Previous work has learned from whole slide images (without patching) by training model modules separately (not end-to-end) (Zhang et al., 2022). Recently, the use of multi-scale patches, including larger patches that provide more spatial context ($4096 \times 4096$ pixels) have been shown to be effective for learning (Chen et al., 2022).

In this work, we perform end-to-end training on gigapixel images (1.024 billion pixels) without distributing the model or data across multiple GPUs. We use a batch size of 1, and a small convolutional neural network that leverages large kernels with high stride. The model is trained to detect tissue from background as a proof of concept. Our training scheme is enabled by a Mac Studio with an M1 Ultra SoC with 128 GB of unified RAM (shared CPU/GPU RAM). A visual depiction can be seen in Figure 1.

## 2. Methods

The data consists of 342 hematoxylin and eosin (H&E) whole slide images acquired under institutional review board (IRB) approval (Vanderbilt IRBs #191738 and #191777). Briefly, 256 images were used for training, 5 for validation, and 81 for testing. The images were converted to grayscale, color inverted, normalized 0 to 1, and cropped to $16000 \times 64000$ pixels. Labels were created by cropping, downsampling by 8, tiling into non-overlapping $1 \times 1$ tiles, thresholding tiles with mean intensity over 230 to 0, conversion to grayscale, thresholding

non-zero pixels to 255, median blurring, erosion, dilation, hole filling, upsampling to the original resolution, re-thresholding values above 0 to 255, and binarizing.

We used a heavily modified U-Net (Ronneberger et al., 2015) with 7 convolutional layers and 4492 parameters. Skip connections used addition rather than concatenation. Early layers learned a downsampling with few large kernels and high stride (no pooling). All training and inference was performed on a Mac Studio with M1 Ultra SoC and 128 GB of unified memory. The model was trained with the Adam optimizer, a learning rate of 1e-3, a batch size of 1, and binary cross-entropy loss. The model weights corresponding to the lowest validation loss were selected for evaluation.

## 3. Results & Discussion

The average time per step (including validation) was 1 minute and 21 seconds. The Apple Activity Monitor application was used to get an accurate read on the unified memory usage for the process. The peak unified memory consumption from the first 8 steps of training was 103.61 GB. The model achieved a Dice score of $0.989 \pm 0.005$ on the testing set. Figure 2 shows a qualitative assessment of model performance.

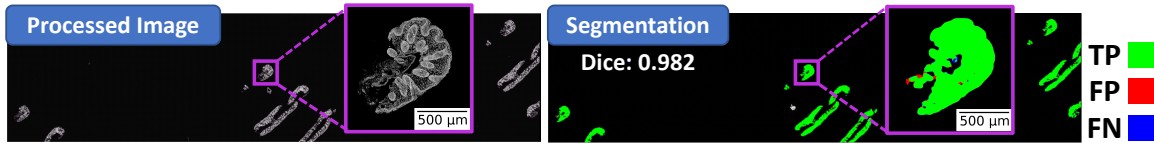

Figure 2: Segmentations with a large amount of true positives led to high Dice scores. This proof of concept demonstrates learnability of gigapixel images using Apple silicon.

In this proof of concept, we have shown that it is possible to perform end-to-end training directly on images of size 1.024 billion pixels, with no patching, using $5000 Apple silicon (Mac Studio with M1 Ultra SoC and 128 GB of unified memory). The peak unified memory usage that was measured for the process was 103.61 GB. During development, this model was unable to run on an NVIDIA RTX A6000 (48 GB, $\sim$ $4500). Other high RAM GPUs, such as the NVIDIA A100 (80 GB, $\sim$ $15000) are not yet widely available (in preview for select regions on Amazon at $40.96/hour for a group of 8). The shared GPU/CPU RAM architecture on the Apple M1 Ultra SoC opens the field for the design of memory-efficient models for gigapixel images at a reasonable price point.

### Acknowledgments

This research was supported by The Leona M. and Harry B. Helmsley Charitable Trust grant G-1903-03793 and G- 2103-05128, NSF CAREER 1452485, NSF 2040462, NSF GRFP Grant No. DGE-1746891, NCRR Grant UL1 RR024975-01 (now at NCATS Grant 2 UL1 TR000445-06), the NIDDK, VA grants I01BX004366 and I01CX002171, VUMC Digestive Disease Research Center supported by NIH grant R01DK135597, P30DK058404, NIH grant T32GM007347, NVIDIA hardware grant, resources of ACCRE at Vanderbilt University.

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
