# OpenReview forum: "Exploring shared memory architectures for end-to-end gigapixel deep learning"
_MIDL.io/2023/Short_Paper_Track — MIDL 2023 Short paper track Poster_

### Official Review · Reviewer_HZXa · 2023-04-20
**Interesting topic with some key questions to clariy**

**Rating:** 6
**Confidence:** 3

**Review:**

Summary
This work investigated doing gigapixel deep learning using the Apple M1 chip, and demonstrated its performance in a gigapixel image segmentation task.

Strengths
(1) Doing deep learning in a super-high resolution of images (especially medical images that are usually available with high resolutions) with related-low-cost is a very promising research direction.
(2) This paper demonstrated that the Apple M1 chip is able to handle gigapixel deep learning with a batch size of 1. And this solution is generally less expensive compared with other hardware configurations.

Weaknesses
(1) The experiment's settings are not very convincing. First, the validation set is too small, only 5 out of 342 total data, this will make the test performance very sensitive to these 5 validation points. Second, batch_size=1 seems not a good choice for deep learning, which can make the SGD very unstable.
(2) Besides, it is worth comparing the gigapixel performance (batch_size=1) with current common practices, like 256*256 pixels (whole image, downsampled) with 64 or 128 batch size. This can give readers much better insights: not only now we "can do" gigapixel deep learning, but also demonstrate the benefit of doing that.
(3) I might have missed it, but is there any novelty in engineering or detailed description in the paper for using the Apple M1 chip for gigapixel deep learning? Can one just deploy the large neural networks on the chip and the chip will handle it automatically?

---

### Official Review · Reviewer_tpea · 2023-04-24
**This paper investigates the potential of shared GPU/CPU memory architectures (particularly on A recently released M1 Ultra systems-on-a-chip) to train deep networks on gigapixel images.**

**Rating:** 7
**Confidence:** 4

**Review:**

The problem of training deep learning models on large-scale image data has been a long-standing problem. The authors tested the performance of a recently released mac product with shared memory architecture on gigapixel images without having to divide them into patches. With a surprisingly low batch size (set as one) on billions of image pixels, the segmentation task achieves a high dice score (>98%). While this is not a technical paper, the findings will be of a high interest to MIDL community.